# One-Pot Decoration of Cupric Oxide on Activated Carbon Fibers Mediated by Polydopamine for Bacterial Growth Inhibition

**DOI:** 10.3390/ma13051158

**Published:** 2020-03-05

**Authors:** Hangil Moon, Young-Chul Lee, Jaehyun Hur

**Affiliations:** 1Department of Chemical and Biological Engineering, Gachon University, Seongnam-si, Gyeonggi-do 13120, Korea; etenelsnow@naver.com; 2Department of BioNano Technology, Gachon University, Seongnam-si, Gyeonggi-do 13120, Korea

**Keywords:** activated carbon fibers, cupric oxide, polydopamine, antimicrobial effect, mechanical property

## Abstract

Despite the widespread application of activated carbon fiber (ACF) filters in air cleaning owing to their high surface area and low price, they have certain limitations in that they facilitate bacterial growth upon prolonged use as ACF filters can provide favorable conditions for bacterial survival. The deposition of cupric oxide (CuO) on ACFs can be an effective way of resolving this problem because CuO can inhibit the proliferation of bacteria owing to its antimicrobial properties. However, finding a new method that allows the simple and uniform coating of CuO on ACF filters is challenging. Here, we demonstrate one-pot CuO deposition mediated by polydopamine (PD) to realize an ACF filter with antimicrobial activity. Scanning electron microscopy (SEM), energy dispersive X-ray spectroscopy (EDS), and X-ray photoelectron spectroscopy (XPS) analyses reveal that CuO and PD are uniformly deposited on the ACF surface. The amount of CuO formed on the ACFs is measured by thermogravimetric analysis (TGA). Finally, the changes in surface area, pressure drop, and antimicrobial activity after coating PD-CuO on the ACFs are evaluated. The use of PD-CuO on the ACFs effectively suppresses the growth of bacteria and enhances the mechanical properties without significantly sacrificing the original characteristics of the ACF filter.

## 1. Introduction

Activated carbon fibers (ACFs) are an activated carbon-based porous fibrous material with high adsorption properties. ACFs are made of various raw materials such as cellulose, phenolic resin, polyacrylonitrile, and coal tar pitch fibers [1,2]. Many different types of ACFs can be produced using different processes that determine the properties including surface area, pore volume, pore diameter, and adsorbed amount of chemicals. Owing to these favorable characteristics, ACF filters have been widely used in air purification, especially for the adsorption of volatile organic compounds and other hazardous gaseous pollutants [3,4,5]. Additionally, ACF has emerged as a promising material for filtering fine dusts which adversely affect human health and have significant negative effects on the respiratory tract [6]. However, unfortunately, the long-term use of ACF filters, especially under humid conditions, allows the growth of deleterious bacteria because ACF filters facilitate bacterial proliferation [7,8]. Therefore, the use of ACF filters may cause secondary problems for humans.

Cupric oxide (CuO) is one of the metal oxides that show antimicrobial activity owing to the release of Cu ions which disrupt the local living environment of bacteria [9,10]. Although a high concentration of CuO nanoparticles is required to achieve the antibacterial effect, compared with other bactericides such as Ag and Cu nanoparticles, CuO is relatively less toxic to the environment and organisms, especially compared with Ag nanoparticles that are the most widely used antibacterial materials [11]. Additionally, CuO is much cheaper than Ag and very stable in ambient conditions.

Many different approaches have been adopted to deposit metals or metal oxides on fibrous surfaces [12]. Electroless plating is a useful method as it can deposit metallic as well as non-metallic materials without the need for electrically conductive substrates [13]. However, it involves time-consuming and cumbersome processes such as cleaning, etching, sensitization, and activation. Particularly, the cleaning process requires strong acids which are harmful to the environment. Electroplating is another method that enables the deposition of metals such as Cu, Au, Ag, and Cr. However, this technique usually requires a more complex experimental setup (requirement of anode, cathode, appropriate pH, temperature, supporting electrolyte, electric power sources, etc.) [14]. Moreover, the substrate should be electrically conductive and only a limited number of materials can be deposited [15].

The deposition of metals or metal oxides mediated by polydopamine (PD) is a simple, environment-friendly, and efficient method as it does not require a complex experimental setup and toxic chemicals. Moreover, there is almost no limitation on the type of substrate (hydrophilicity/hydrophobicity, conductivity, surface roughness, etc.) due to the strong adhesion of PD on the substrate, which mediates the formation of metals or metal oxides [16,17]. This is achieved by the catechol and amine groups in dopamine which allow a strong adhesion as well as reduction during the polymerization. The reduction potential of −530 mV (versus normal hydrogen electrode (NHE)) of the catechol group allows the reduction of most metal precursors to form metals or metal oxides [18]. This reduction is generally carried out in a sequential manner; that is, the substrate is coated by PD followed by the reduction of the metal precursor. However, recently, one-pot coating of PD and metal has been demonstrated in which the metal (Au or Ag) is reduced during the polymerization of dopamine [19,20,21]. This approach is simpler and less time-consuming than the conventional sequential method. Son et al. demonstrated hybrid one-pot coating of PD-Ag on electrospun poly(vinyl alcohol) nanofibers for antibacterial filters [19]. Yoon et al. prepared Ag nanoparticle-deposited ACF using electroless deposition to avoid bacterial contamination [8]. Ren et al. showed the feasibility of the antimicrobial activity of pure Cu and Cu_2_O nanoparticles prepared using thermal plasma technology [22]. Despite these efforts, none of previous studies have demonstrated the CuO-coated ACF using a simple one-pot PD-mediated deposition process.

During this work, we demonstrate the one-pot CuO coating of an ACF filter mediated by PD to obtain an ACF filter with antimicrobial activity. The CuO nanoparticles homogeneously and strongly cover the ACFs during the polymerization of dopamine. The CuO-coated ACF filter exhibits nearly perfect antibacterial activity (~100%) against both gram-positive and gram-negative bacteria. Although the surface area is reduced to some extent after PD-CuO coating, the pressure drop is almost negligible. The mechanical stability of the ACFs is even improved owing to the organic (PD)-inorganic (CuO) hybrid coating on the ACFs, which is beneficial for application as a filter.

## 2. Materials and Methods

### 2.1. Materials 

ACFs (STF-1600, 100% viscose base, commonly known as artificial cotton) were provided by Sutong Carbon Fiber Co., Ltd. (Jiangsu, China). Dopamine hydrochloride (product #: H8502, CAS: 62−31−7), sodium acetate (CH_3_COONa, product #: 32319, CAS:127-09-3), Tween^®^ 20 (product #: P9416, CAS:9005-64-5), copper sulfate pentahydrate (CuSO_4_∙5H2O, product #: 31293, CAS:7758-99-8), and phosphate-buffered saline (PBS, product #: P4417, PCode:1002852269) were purchased from Sigma Aldrich (St. Louis, MO, USA). BBL^TM^ eosin methylene blue agar, modified (Holt–Harris and Teague) was bought from Becton, Dickinson and Company (Franklin Lakes, NJ, USA). Luria Bertani (LB) broth (Miller’s) solution (10 g of tryptone, 5 g of yeast extract, and 10 g of NaCl dissolved in 1 L of deionized water) was provided by Genomicbase (Seoul, South Korea). Agar powder was supplied by Duksan Chemicals Co. Ltd. (Ansan, South Korea).

### 2.2. Material Characterization

The elemental compositions of the samples were determined by X-ray photoelectron spectroscopy (XPS) using a monochromatic Al Kα X-ray source (1486.6 eV) and an AXIS ultra-delay line detector (DLD) (Kratos, Manchester, UK). Thermogravimetric analysis (TGA) was performed under an air atmosphere from 20 °C to 800 °C at a ramping rate of 10 °C/min using an SDT Q-600 analyzer (TA Instrument, New Castle, DE, USA). The microstructure of the sample was observed by scanning electron microscopy (SEM, Hitachi S-4700, Tokyo, Japan) and transmission electron microscopy (TEM, JEM-2100F, JEOL, Tokyo, Japan). Regarding TEM sample preparation, ACF@PD-CuO powder was milled using a planetary mechanical milling machine (Pulverisette 5, Fritsch, Idar-Oberstein, Germany) at a speed of 300 rpm for 2 h. The functional groups of the ACFs were analyzed by Fourier transform infrared (FTIR) spectroscopy (Vertex 80v, Bruker, Billerica, MA, USA) in the range of 4000–400 cm^−1^. Brunauer–Emmett–Teller (BET) surface area analysis was performed to measure the nitrogen adsorption isotherms using an ASAP2020 system (Micromeritics, Norcross, GA, USA).

### 2.3. Preparation of activated carbon fiber (ACF) Coated with Polydopamine and Copper(II) Oxide (ACF@PD-CuO)

The ACFs were washed with ethanol and distilled water at least three times, and then dried overnight in an oven at 55 °C. Separately, copper sulfate (25 mM) and sodium acetate (50 mM) were dissolved in deionized water (the pH at this condition was 5.0) at 55 °C for 1 h. Dopamine hydrochloride (4 g/L) was added to this solution (the color was changed from blue to brown immediately after the addition of dopamine hydrochloride) followed by the immersion of dried ACFs with continuous stirring at a speed of 130 rpm for 12 h at 55 °C. During this process, the polymerization of dopamine and reduction of Cu ions occurred simultaneously to form ACF@PD-CuO. After completion of the reaction, ACF@PD-CuO was washed with distilled water several times and dried at 55 °C overnight. ACF@PD was separately prepared following the same method, except for the use of a different buffer solution (Tris buffer at pH 8.5) without the addition of copper sulfate and sodium acetate. 

### 2.4. Antimicrobial Test

*Escherichia coli* (*E. coli* ATCC 25922) and *Staphylococcus aureus* (*S. aureus* ATCC 29213) were chosen as the Gram-negative and Gram-positive bacteria, respectively. Each bacterial strain (0.1 mL) was inoculated in 10 mL of nutrient broth and incubated at 37 °C for 24 h. The nutrient broth for making agar was sterilized at 120 °C in an autoclave for 15 min. Then, 20 mL of nutrient broth was dispensed in a Petri dish (diameter of 9 cm) and dried. The absorbance at 600 nm was used to determine the degree of incubation. The activated carbon fibers (ACFs) coated with polydopamine-cupric oxide (PD-CuO) (0.4 g) were inoculated with 1 mL of bacterial culture and incubated at 37 °C for 24 h. Then, 20 mL of physiological saline solution (PSS) with NaCl and Tween^®^ 20 was added to the ACFs followed by vortexing. Subsequently, the nutrient agar was inoculated with 20 μL of PSS, which contained the living bacteria from the ACFs. The nutrient agar was spread and incubated at 37 °C for 24 h. The antibacterial activity was measured by counting the colonies in the nutrient agar. The antimicrobial test was performed based on 10^6^–10^7^ colony forming units.

### 2.5. Pressure Drop Measurement 

The pressure drops of the air flow across the activated carbon fibers (ACF)-based filters were measured using a differential pressure-measuring instrument (Testo 510i, Testo, Germany) in an acrylic duct (length: 42 cm, width: 8 cm, and height: 8 cm). Concerning this setup, a filter (9 cm × 9 cm) was inserted into a custom-made frame in the center of the duct. The velocity of airflow was controlled using a vacuum pump (JBA00117, Jungwoo, South Korea).

### 2.6. Mechanical Property Measurement

The mechanical properties of the ACF-based filters were evaluated using a tensile testing machine (model 3345, Instron, Norwood, MA, USA). The tests were performed on the samples with a gauge length of 50 mm at a speed of 50 mm/min three times for each type of sample at 20 ± 2 °C and a relative humidity of 65 ± 4%.

## 3. Results and Discussion

Figure 1 schematically describes the mechanism of simultaneous deposition of polydopamine (PD) and cupric oxide (CuO) on the activated carbon fibers (ACF) surface. The catechol group present in dopamine has multiple functionalities: i) adhesive property, ii) reduction capability for Cu ions, iii) polymerization of dopamine (formation of PD) through oxidization in a basic aqueous environment, and iv) mechanical reinforcement of the ACFs due to the simultaneous polymerization and reduction of Cu ions. The mussel-inspired adhesion property of dopamine has been widely reported in the literature [23,24]. The adhesive characteristics of PD can be predominantly utilized to coat desired surfaces, regardless of the substrate surface property (i.e., hydrophilicity or hydrophobicity). Despite the low surface energy of ACFs (32−45 mJ/m^2^), PD can strongly adhere on the ACFs through the catechol groups in PD and can simultaneously reduce the Cu ions to form CuO [19,20]. The reduction ability of dopamine stems from the oxidation of the catechol group, which releases two electrons and protons, followed by polymerization to form PD. The mild redox potential (−530 mV versus normal hydrogen electrode (NHE)) of the catechol group enables the reduction of Cu ions (the reduction potential required to reduce Cu^2+^ is 340 mV) [18]. Additionally, the PD layer formed with CuO enhances the mechanical property of the substrate owing to the inorganic coating layer (CuO), which will be advantageous for the ACFs in filter applications. More importantly, the formation of CuO mediated by PD imparts antimicrobial properties to the ACF filter.

The morphology of the ACFs coated with PD-CuO (ACF@PD-CuO) was observed by scanning electron microscopy (SEM, Figure 2a). While the pristine ACFs showed a very smooth surface, ACF@PD-CuO exhibited many rugged surfaces that are regarded as PD aggregates. This is confirmed by the separate SEM imaging of ACF@PD without CuO where the similar aggregates were observed (Appendix A). These aggregated parts have been observed in previous studies in which bare PD was coated on other substrates such as polymer nanofibers, nanoparticles, and films [19,25,26]. The aggregated parts may have formed due to the internal covalent bonding between PDs that resulted from the reaction between quinones and the amine groups in the catechol quinones (dopamines), leading to the formation of indole compounds [19]. Another possible reason for the aggregated particle formation is the non-covalent bonding, such as hydrogen bonding, between catechol quinones [27]. More aggregated parts were observed with an increase in coating time, which indicated continuous reactions between the internal dopamine molecules.

However, the uniform distribution of Cu atoms observed in the energy dispersive spectroscopy (EDS) color mapping images indicates that the smooth regions in ACF@PD-CuO are covered by CuO (Figure 2b). The homogeneous Cu distribution in ACF@PD-CuO-4h (Figure 2b) suggests that PD-CuO fully covered the ACF surface after 4 h of coating. Note that it is difficult to acquire the elemental mapping image for nitrogen due to the similar peak positions of nitrogen and carbon atoms in EDS analysis, which resulted in the negligible peaks from nitrogen due to the dominating peaks from carbon [28]. The further deposition of PD-CuO on the ACFs increased the thickness of PD-CuO until 12 h, as indicated by the increase in the weight percentage of Cu in ACF@PD-CuO with increasing coating time: 7.1, 9.1, and 10.9% for 4, 8, and 12 h of coating, respectively (EDS analysis shown in Figure 2c and Appendix A). The thicknesses of PD-CuO measured from the cross-sectional SEM imaging were ~15, ~23, and ~27 nm for 4, 8, and 12 h of coating, respectively as shown in Appendix A. Despite the steady increase in coated PD-CuO, the morphological change seemed to be insignificant due to the extremely low thickness of PD-CuO on the ACFs (~27 nm, Appendix A) even after 12 h.

Due to the limitation of EDS analysis that can provide only local information of atomic content (spot analysis), more general information regarding the amount of coated constituents (PD and CuO) were obtained by thermogravimetric analysis (TGA). Figure 3 shows the thermal decomposition profiles of pristine ACF, ACF@PD, and ACF@PD-CuO-12h in air. While pristine ACF exhibited a sharp weight loss at ~500 °C due to the decomposition of carbon–carbon bonds, ACF@PD showed a three-step weight loss: i) evaporation of remaining moisture until ~90 °C, ii) gradual decomposition of PD starting from ~100 °C (region 1 and 2), and iii) decomposition of both PD and ACF after ~500 °C (region 3). Occurring at ~650 °C, almost no residual ACF or PD remained for ACF@PD [29,30]. However, ACF@PD-CuO exhibited a very different thermal decomposition trend compared with that of the pristine ACFs or ACF@PD. ACF@PD-CuO showed a trend similar to that of ACF@PD up to ~250 °C (region 1), but then displayed a sharp weight loss (region 2). This is because the decomposition of carbon in the ACFs and PD was accelerated by CuO, which acted as a catalyst for the oxidation of carbon [2,31]. Even if this thermal behavior of ACF@PD-CuO is different from ACF or ACF@PD, it does not influence the stability of ACF@PD-CuO because this filter is not usually used in the temperatures above 250 °C in normal situations. The weight percent of the residue after the complete decomposition of ACF and PD was ~9.3%, which was considered to be the amount of CuO deposited on the ACFs.

The surface chemical composition of the deposited PD-CuO was analyzed by X-ray photoelectron spectroscopy (XPS, Figure 4). Shown in Figure 4b, ACF@PD-CuO displays a distinct N_1s_ peak at 400.3 eV, which is indicative of C–N bonding in PD [32,33]. The high-resolution scan of the C_1s_ peak shows the presence of C–N bonding at 286 eV, further confirming the formation of PD (Figure 4c) [32]. Using O_1s_ analysis, the deconvoluted peak at 530.3 eV confirms the presence of CuO and the secondary peak at 531.8 eV is attributed to the surface hydroxide, which mainly originates from the catechol groups in PD (Figure 4d) [34]. A close view of the high-resolution Cu_2p_ scan reveals a peak at 933.6 eV with a satellite peak at 941.2 eV for Cu_2p3/2_, as well as peaks at 953.8 eV and 961.6 eV for Cu_2p1/2_, which correspond to the CuO phase (Figure 4e) [35,36]. Viewing the separate TEM analysis, CuO was revealed to exist in atomic scale rather than to be nanoparticles (Appendix A). The formation of PD and CuO was further confirmed by Fourier transform infrared (FTIR) analysis (Figure 5). The peaks in the frequency ranges of 3200–3700 cm^−1^ (region 1, O-H stretching of the hydroxyl group) and 1360–1580 cm^−1^ (region 2, N-H stretching of secondary amine) significantly increased for ACF@PD-CuO compared with those in the pristine ACFs owing to the presence of PD on the ACFs [37,38,39]. Additionally, ACF@PD-CuO shows a distinct peak at 1640 cm^−1^ corresponding to the stretching vibration of CuO, which further confirms the presence of CuO in ACF@PD-CuO [40,41].

The change in the surface area of the ACF-based filter is important to determine whether the adsorptive characteristics of the filter can be effectively maintained after PD-CuO deposition. Figure 6 shows the surface areas of the ACFs and ACF@PD-CuO at different coating times determined from the nitrogen adsorption isotherms obtained by the BET method. The high surface area of the pristine ACFs (1098.674 m^2^/g) decreased with increasing deposition time due to the blockage of the micropores in the ACFs by PD and CuO. However, this blockage does not seem to be perfect considering the fact that once micropores are completely covered by PD and CuO, the surface area of ACF should be in the order of ~1 m^2^/g from the given fiber diameter (~2 μm) and density (~2000 kg/m^3^) of ACF. Thus, it is possible that the micropores could be partially covered by PD and CuO or some cracks in PD-CuO could have been generated during the drying process which allowed N_2_ to penetrate through the free surfaces during the BET measurements. Nevertheless, the degree of surface area decrease reduced after 2 h of deposition and almost no significant change was observed in the surface area after 6 h. This reflects the initial coating of PD-CuO mostly covered the micropores, and subsequent deposition only increased the thickness of the PD-CuO layer, which does not significantly affect the surface area of the ACF. The decrease in surface area from 2 h to 12 h was only 19.35%, for example. This is due to two reasons: i) the thickness increase is insignificant (e.g., tens of nanometers even after 12 h, as shown in Appendix A), and ii) some of the deposited PD-CuO was used in the formation of aggregates, which are sporadically observed on the ACFs (Figure 2a). This suggests that a longer coating time of PD-CuO can improve the antimicrobial activity owing to an increase in CuO without significantly sacrificing the surface area of the ACFs.

Figure 7 shows the change in pressure drop across the ACFs and ACF@PD-CuO with different coating times as a function of airflow velocity in the range of 0.1–0.3 m/s. Regarding the pristine ACFs, the pressure drop increased with the increasing flow rate due to the elevated degree of flow resistance through the filter: 5, 21, and 57 Pa for 0.1, 0.2, and 0.3 m/s, respectively. However, the pressure drop was not greatly affected by PD-CuO deposition, irrespective of the coating time. Considering a certain level of decrease in surface area after PD-CuO coating, these results indicate that the pressure drop is not directly associated with the surface area. Rather, the insignificant change in pressure drop after PD-CuO coating is probably due to the deposition of a thin layer (a few tens of nanometers after 12 h of coating). The slight change in pressure drop after the PD-CuO coating of the ACFs should be beneficial to its application as a filter.

Figure 8a and b show the investigation of the antibacterial effect of developed samples against *E. coli* and *S. aureus*, respectively. Visibly, the pristine ACFs facilitate the growth of bacteria. This is because bacteria can easily adhere to the biocompatible ACF filter support, which provides a favorable environment for the survival of bacteria. After repeated experiments, ~20–30% more bacteria were found in the nutrient agar for the pristine ACFs compared with the nutrient agar without the pristine ACFs (No ACF) for both kinds of bacteria. However, in the case of ACF@PD-CuO-12h, 99.9% of antimicrobial activities against both Gram-positive and Gram-negative bacteria were obtained. The antimicrobial activity was maintained as ~100% after a coating time of 4 h. We also confirmed the absence of any viable bacteria on ACF@PD-CuO after the antimicrobial test via ex-situ SEM measurements (Appendix A). It has been reported that the antibacterial activity against Gram-negative bacteria is lower than that against Gram-positive bacteria [8,42]. The sensitivity of Gram-negative bacteria is lower than that of Gram-positive bacteria due to the presence of lipopolysaccharide molecules which act as a barrier to Gram-negative bacteria [8,43]. However, ACF@PD-Cu-12h retained a sufficiently high antimicrobial activity and overcame this limitation. Moreover, since pure PD is ineffective against bacterial proliferation despite the presence of the amine groups of dopamine [16,19,44], the combination of PD and CuO is highly effective in preventing the growth of both Gram-positive and Gram-negative bacteria.

The mechanical property of the filter is one of the important factors in its long-term use in a harsh environment. Figure 9 and Table 1 show the tensile properties of the pristine ACFs and ACF@PD-CuO. ACF@PD-CuO-12h exhibited much improved mechanical properties compared with those of the pristine ACFs. The deposition of PD-CuO on the ACFs resulted in ~10- and ~5-fold increases in Young’s modulus and tensile strength, respectively, compared with the pristine ACFs. The enhanced mechanical property is mainly attributed to the use of inorganic CuO although the elongation and tensile strain decreased after PD-CuO deposition due to the rigid nature of CuO. Additionally, considering that a number of intrinsic defects in the pristine ACFs may serve as breakpoints upon mechanical deformation during use, the introduction of PD is beneficial as it not only fills these defect sites, but also enhances the mechanical properties of the ACFs [45].

## 4. Conclusion

To summarize, we demonstrated the fabrication of activated carbon fibers@polydopamine-cupric oxide (ACF@PD-CuO) for application as a filter with antimicrobial activity. PD and CuO were deposited simultaneously via the polymerization of dopamine and reduction of Cu ions without involving any toxic chemical processes. Compared with previous approaches to prepare the antibacterial filters which involved environmentally toxic Ag nanoparticles or difficult deposition processes, our method is much simpler, cost-effective, and environmentally friendly. The morphology and chemical properties of ACF@PD-CuO analyzed by scanning electron microscopy (SEM), transmission electron microscopy (TEM), energy dispersive spectroscopy (EDS), X-ray photoelectron spectroscopy (XPS), thermogravimetric analysis (TGA), and Fourier transform infrared (FTIR) spectroscopy revealed that a PD-CuO layer was successfully deposited on the ACFs. The prepared ACF@PD-CuO exhibited perfect antimicrobial activities against both Gram-positive and Gram-negative bacteria. After the introduction of PD-CuO, although the surface area (i.e., filtration efficiency) slightly decreased, the level of pressure drop remained almost unchanged. Moreover, the mechanical properties of the ACF were significantly enhanced after coating with PD-CuO, which served as reinforcements for the pristine ACFs. Overall, ACF@PD-CuO prepared by the simple one-pot coating method is a promising filter that can effectively inhibit bacterial growth without losing its various original merits.

## Figures and Tables

**Figure 1 materials-13-01158-f001:**
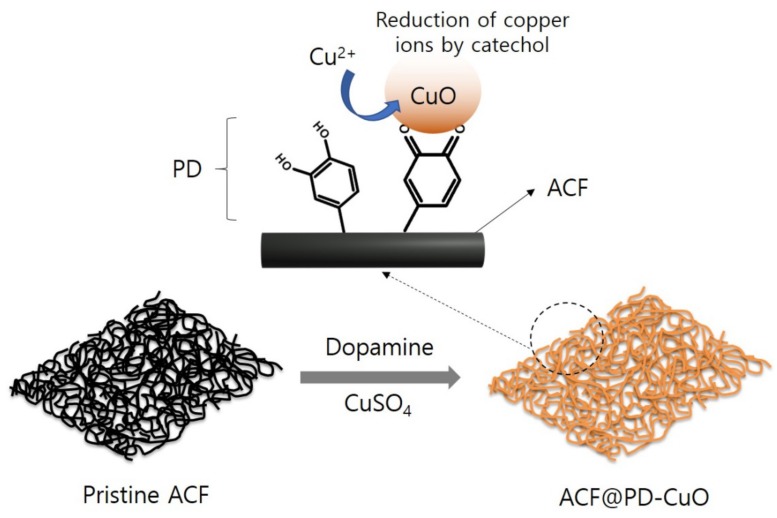
Schematic illustration of the mechanism of simultaneous deposition of polydopamine (PD) and CuO on the activated carbon fiber (ACF) surface.

**Figure 2 materials-13-01158-f002:**
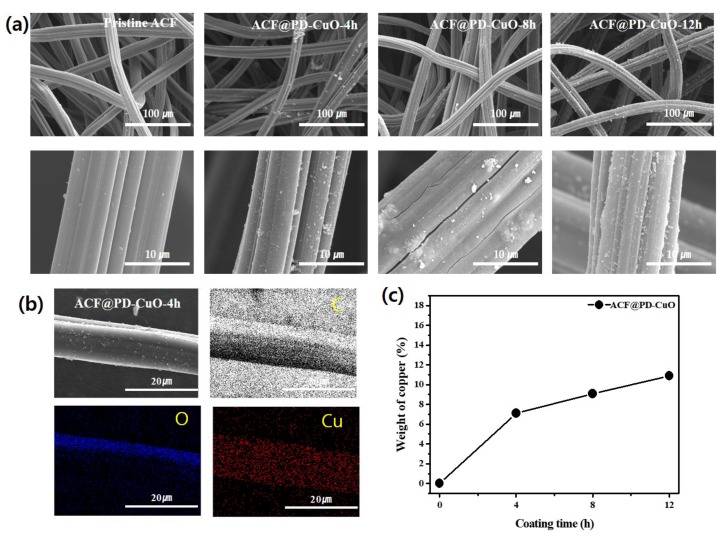
(**a**) Scanning electron miscoscopy (SEM) images of pristine activated carbon fibers (ACF) and ACF@polydopamine-cupric oxide (PD-CuO) with a coating time of 4, 8, and 12 h, respectively, (**b**) energy dispersive spectroscopy (EDS) mapping images of C, O, and Cu of ACF@PD-CuO-4h, and (**c**) copper content as a function of coating time obtained from EDS analysis (Appendix A).

**Figure 3 materials-13-01158-f003:**
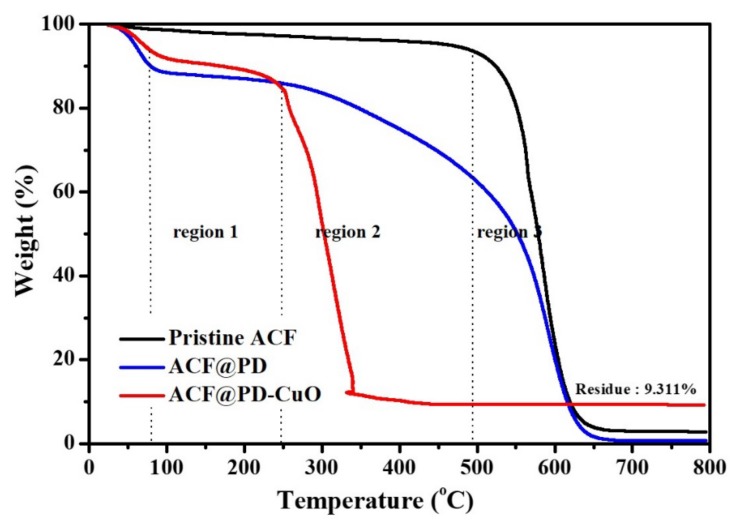
TGA curves of pristine ACF, ACF@PD, and ACF@PD-CuO.

**Figure 4 materials-13-01158-f004:**
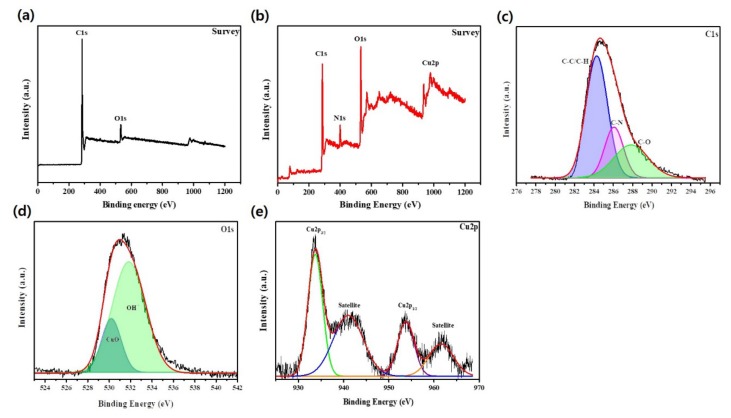
X-ray photoelectron spectroscopy (XPS) survey spectra of (**a**) pristine activated carbon fibers (ACF) and (**b**) ACF@ polydopamine-cupric oxide (PD-CuO)-12h, (**c–e**) deconvolution of peaks for C_1s_, O_1s_, and Cu_2p_, respectively.

**Figure 5 materials-13-01158-f005:**
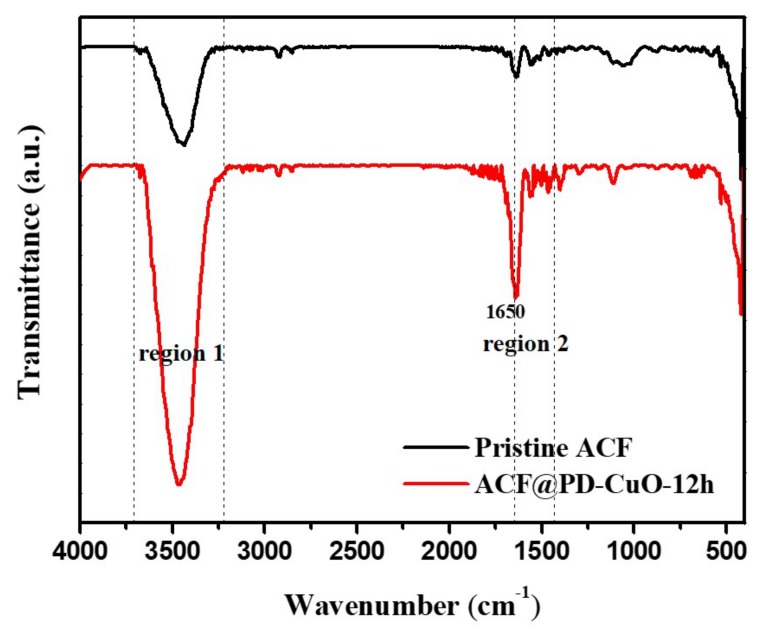
Fourier transform infrared (FTIR) spectra of pristine activated carbon fibers (ACF) and ACF@polydopamine-cupric oxide (PD-CuO)-12h.

**Figure 6 materials-13-01158-f006:**
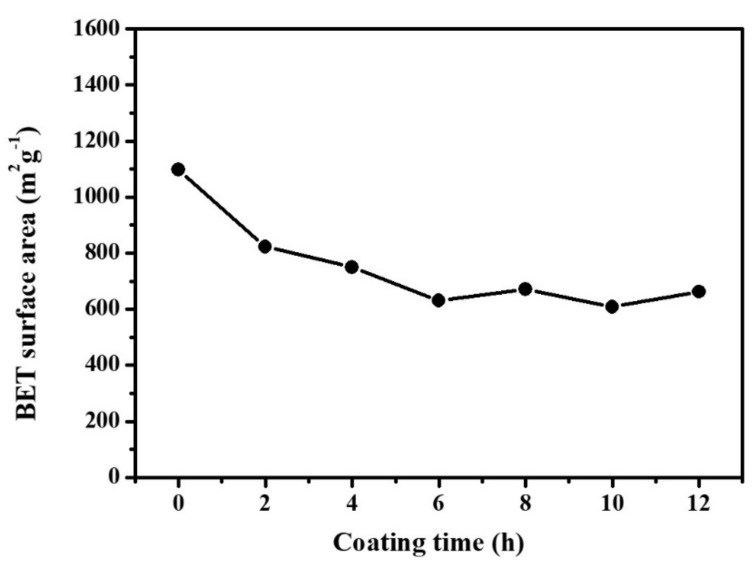
Brunauer–Emmett–Teller (BET) surface area of the pristine activated carbon fibers (ACF) and ACF@polydopamine-cupric oxide (PD-CuO) with different coating times.

**Figure 7 materials-13-01158-f007:**
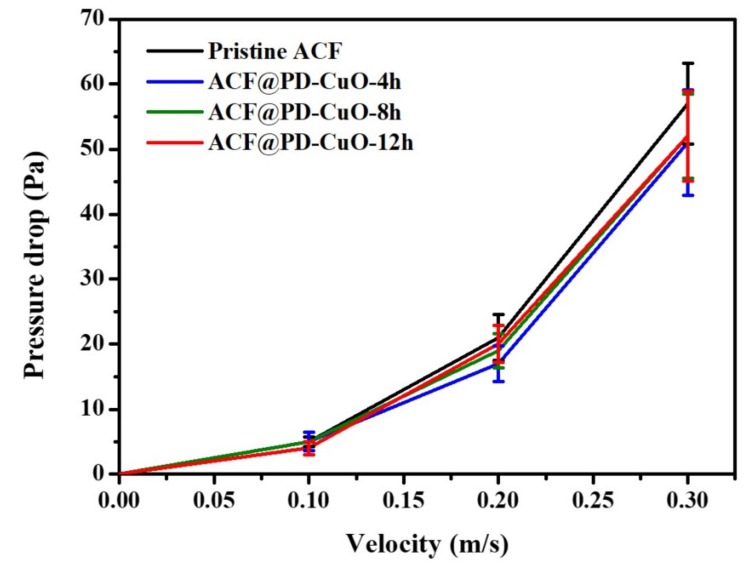
Pressure drop of pristine activated carbon fibers (ACF) and ACF@polydopamine-cupric oxide (PD-CuO) filters as a function of air velocity with different coating times (4, 8, and 12 h).

**Figure 8 materials-13-01158-f008:**
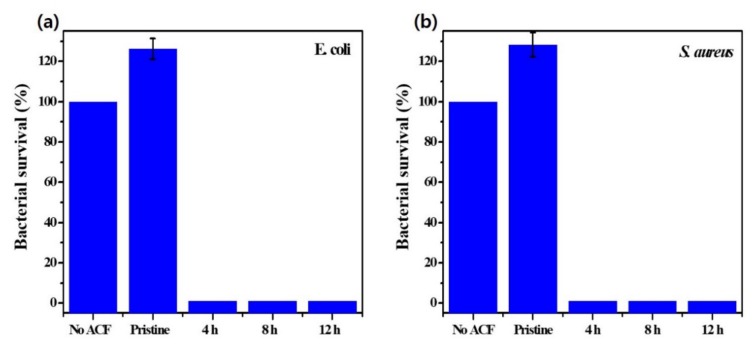
Antimicrobial effects of the activated carbon fibers (ACF) and ACF@polydopamine-cupric oxide (PD-CuO) with different coating time against (**a**) *E. coli* and (**b**) *S. aureus.*

**Figure 9 materials-13-01158-f009:**
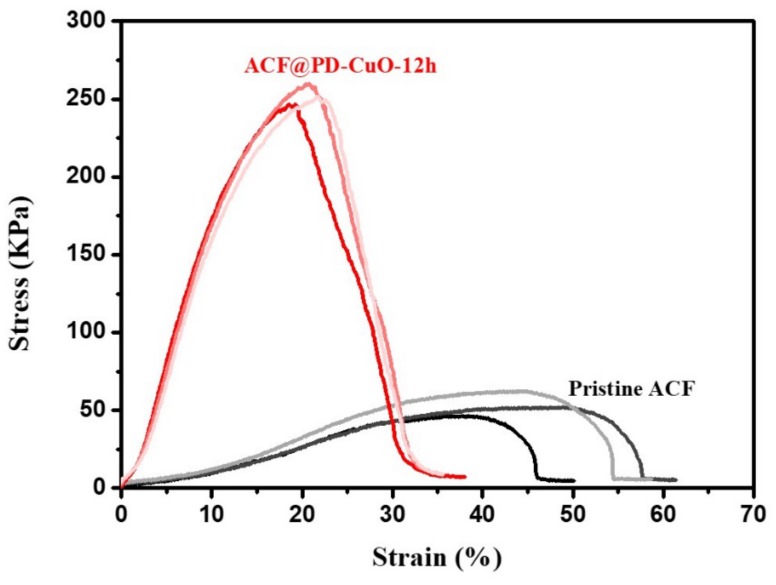
Stress-strain curve of pristine activated carbon fibers (ACF) and ACF@polydopamine-cupric oxide (PD-CuO)-12h.

**Table 1 materials-13-01158-t001:** Physical properties of pristine activated carbon fibers (ACF) and ACF@polydopamine-cupric oxide (PD-CuO). All the values are averages of 3 measurements.

	Young’s Modulus (MPa)	Tensile Strength (kPa)	Elongation (%)	Tensile Strain(%)
Pristine ACF	0.21 ± 0.03	53.73 ± 7.97	43.20 ± 5.46	56.74 ± 5.93
ACF@PD-CuO-12h	2.09 ± 0.06	252.82 ± 6.61	20.63 ± 1.26	36.35 ± 1.49

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
