# Peer review of "One-Pot Decoration of Cupric Oxide on Activated Carbon Fibers Mediated by Polydopamine for Bacterial Growth Inhibition"

_materials, 2020, doi:10.3390/ma13051158_

Round 1

Reviewer 1 Report

Dependance of  SSABET  on the coating time as depicted on the Fig. 6. shows stagnation in further decline since 6h at the value around 650 m2/g. This decline is explained in this work such as „due to the blockage of the micropores in the ACFs by PD and CuO“ as we can read in their paper:

„Rows 226-232 "The high surface area of the pristine ACFs (1098.674 m2/g) decreased with increasing deposition time due to the blockage of the micropores in the ACFs by PD and CuO. However, the degree of surface area decrease reduced after 2 h of deposition and almost no significant change was observed in the surface area after 6 h. This reflects the initial coating of PD-CuO mostly covered the micropores, and subsequent deposition only increased the thickness of the PD-CuO layer, which does not significantly affect the surface area of the ACF."

If the surface of the coated fiber was completely closed without pores, then the measured SSABET would be determined only by its outer geometric surface of the fibers. With a diamater of one fiber about 2µm according to Fig. 2 (a) and taking into account only the tapped density of pulverized activated carbon ρap.den ≈ 2000 kg/m3, the SBET would reach a size of about 1 m2/g.

SSABET = 2πrlr2lρap.den = 2/rρap.den = 2/(10-6·2·106) ≈ 1 m2/g.

This value is in significant contrast to the measured SBET = 650 m2/g. The hypothesis about „BLOCKAGE OF THE MICROPORES“ should therefore be further examined and explained mentioned disproportion. One possible explanation could be that the outer surface of the fiber is not completely coated and N2 molecules can penetrate through the free surfaces during BET measurements. These sites, blocked before coating, could be identified as the internal contact of the fibers in the bundle and the inner fibers themselves. Cracks in the coating were very likely to arise only in the dry state and allowed measurement gas to penetrate inside the bundle.

Reviewer 2 Report

The manuscript is interesting and it could be published after a minor revision (as indicated by me in the revised paper). The results regarding the antibacterial activity of developed materials are very impressive! The authors must point out (in Introduction and in Conclusion) the originality and novelty of their research (compared to other studies in the scientific literature).

I mentioned in text some suggestions:

-The Latin words should be written in italic.
-The measurement units “l”, “ml” or “µl” must be written “L”, “mL” and “µL”, respectively.
-Attention to the reference [24]; it must be replaced by other reference, because it is a study about island carbon coating on nitrogen-activated polyurethane surface, and not about the reduction of copper ions. Then, the section References must be updated accordingly.
-Authors must replace the word “substances” with “surfaces”, in the Lines: 148 and 163.
-Authors must specify the coating time for the samples: ACF@PD-CuO in Figures, Tables and in text.

etc.

Authors must specify if they obtained CuONPs or not, and also if they obtained PD-CuO nanoparticles.

I have indicated some comments and corrections in the attached pdf text and in the supplementary material. Authors must pay attention to the words/paragraphs yellow highlighted in text.

Reviewer 3 Report

The paper describe a method of coating active carbon filters (ACF) in order to get them with antibacterial  activity. 

The method involves coating through a reaction with polydopamine and CuO. 

The paper is interesting an is worth to be published. Ihave only some suggestions to the authors:

Line 163-164. The authors stated that SEM analysis of ACF coated with PD-CuO shows aggregates on their surface that are not present on plain ACF. In my opinion these aggregates should be characterized in order to define if they are PD aggregates. Some PD characterization especially referred to aggregates formation, can be found in the literature, i.e. "A.De Trizio, P.Srisuk, R.R. Costa, A. G. Fraga, T. Modena, I. Genta, R. Dorati, J.Pedrosa, B. Conti, V. M. Correlo, R.L. Reis, Natural based eumelanin nanoparticles functionalization and preliminary evaluation as carrier for gentamicin. Reactive and Functional Polymers, 2017, 114: 38-48".

Line 183-184. The authors should measure the thickness of filters after PD-CuO coating.

Line 198. The authors write that decomposiiont of ACF and PD was accelerated by CUO. This means that PD-CuO coating makes ACF less stable? The authors should better discuss this point which can be an important issue.
